# Urban Dominant Trees Followed the Optimal Partitioning Theory and Increased Root Biomass Allocation and Nutrient Uptake under Elevated Nitrogen Deposition

Qinze Zhang, Jiyou Zhu, Jiaan Liang, Meiyang Li, Shuo Huang and Hongyuan Li *

College of Environmental Science and Engineering, Nankai University, Tianjin 300350, China;
zhangqinze1999@163.com (Q.Z.); zhujy@nankai.edu.cn (J.Z.); oneand@mail.nankai.edu.cn (J.L.);
limeiyangal@163.com (M.L.); 2111429@mail.nankai.edu.cn (S.H.)
* Correspondence: eialee@nankai.edu.cn; Tel.: +86-22-2350-6446

**Abstract:** Nitrogen (N) is one of the limiting nutrients for plant growth and metabolism in terrestrial ecosystems. Numerous studies have explored the effects of N addition on the eco-physiological traits and biomass production of plants, but the underlying mechanism of how N deposition influences biomass allocation patterns remains controversial, especially for urban greening trees. A greenhouse experiment was conducted for 7 months, using two dominant tree species of urban streets in North China, including the coniferous tree species *Pinus tabuliformis* and the broadleaved tree *Fraxinus chinensis*, under three levels of N addition: ambient, low N addition, and high N addition (0, 3.5, and 10.5 gN m$^{-2}$ year$^{-1}$). The plant growth, biomass distribution, functional traits, and soil nutrient properties of the two trees were determined. Overall, N addition had positive effects on the aboveground and belowground biomass of *P. tabuliformis*, which also shifted its functional traits to an acquisitive strategy, while *F. chinensis* only increased root biomass distribution and fast traits as N increased. Furthermore, N supply increased the soil N and phosphorus availability of both trees and improved their root nutrient uptake capacity, resulting in an increase in their root–shoot ratio. Optimal partitioning theory could better explain why trees would invest more resources in roots, changing root structure and nutrient uptake, thus increasing root biomass allocation to adapt to a resource-poor environment. These findings highlight the importance of plant functional traits in driving the responses of biomass allocation to environmental changes for urban greening dominant tree species and could help to come up with new tree growth strategies in silvicultural practice for urban green space.

**Keywords:** nitrogen deposition; urban greening trees; optimal partitioning theory; biomass allocation; functional traits

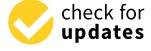



## 1. Introduction

Nitrogen (N), as a primary element in terrestrial ecosystems, controls plant growth and primary productivity [1,2]. Generally, elevated N input increases soil N availability, promoting leaf photosynthesis and root growth, as well as contributing to plant biomass production [3,4]. However, with the increasing frequency of human activities, global atmospheric N deposition has increased rapidly and become a focus of attention [5]. Excessive N deposition causes "N saturation" of the ecosystem, leading to soil acidification and nutrient imbalances, which potentially threaten plant photosynthetic capacity and growth [6]. As for China, the N deposition rate has more than doubled over the past 30 years, especially in urban ecosystems with intensive human activity [7]. Urban greening tree species, as an indispensable part of the urban ecosystems, play important roles in alleviating urban environmental pressure and providing ecosystem services, which have also been affected by global climate change [8]. Thus, it is crucial to explore the response of different greening

trees to N deposition, which includes selecting suitable species, improving the functional urban green space, and creating multiple ecosystem service values.

Plant biomass allocation reflects plant evolution strategies for resource acquisition, growth patterns, and competitive capacity [9,10]. In general, biomass allocation patterns have strong variability due to species specificity and their growth environment [11]. Previous studies have demonstrated that N deposition could change plant biomass production and allocation [10]. Ratio-based optimal partitioning theory, as a plant biomass allocation strategy, shows that plants allocate more biomass proportionally to capture limiting resources in response to environmental fluctuation [9,12], which can explain the effect of N on biomass allocation in the short term [13]. For example, N input could increase the root mass fraction of invasive *Solidago canadensis* to better access nutrients in resource-poor soil [14]. However, it is unclear how N deposition affects the biomass allocation of different tree species, and whether optimal allocation theory is suitable for urban green space remains to be explored.

Furthermore, plant functional traits can indicate the capacity of plants to capture resources and adapt to environmental changes, considered as the potential covariates necessary to understand biomass allocation [15]. Generally, according to the plant economics spectrum strategy, trees with acquisitive traits like a high specific leaf area and root length have easy access to resources and exhibit fast growth rates, while conservative traits such as leaf thickness and root tissue density tend to select the strategy of slow resource acquisition and efficiently preserve resources [16,17]. It has been noted that N addition could modify soil nutrient availability, leading to a transformation of plant functional traits to acquisitive strategies and the ability to obtain resources more quickly [18]. However, the responses of aboveground and belowground traits to N deposition may be different due to different environmental selection pressures [19]. It is necessary to explore the effects of N addition on trees' aboveground and belowground functional traits and how this variation can finally reflect their biomass allocation, so as to fully understand the response mechanisms of urban greening tree species to environmental changes.

Influenced by rapid economic development, urban green spaces receive a higher level of N deposition compared to natural forest ecosystems, especially in North China [20]. However, few studies have explored the underlying mechanism of N inputs on the biomass allocation patterns of different urban greening trees. Understanding the N availability-mediated growth strategy of hardwoods and conifers can provide support for tree species collocation in urban forestry. *Pinus tabuliformis* and *Fraxinus chinensis*, as the dominant coniferous and deciduous trees of urban green spaces in North China, have important ecological and economic value [21]. In this study, seedlings of the two trees were selected and transplanted into a boreal greenhouse and given three N addition treatments. To evaluate the response of tree growth patterns, tree biomass, functional traits, and soil nutrient contents were measured at the end of the cultivation. Specifically, the following hypotheses were put forward: (1) N addition promotes the growth of *P. tabuliformis* and *F. chinensis*, which follows optimal partitioning and provides more resources for root biomass allocation; (2) the variation in tree biomass allocation is also reflected in functional traits, where aboveground and belowground traits choose opposite strategies; (3) N also changes the soil and tree nutrient properties, which can drive a shift in biomass allocation.

## 2. Materials and Methods

### 2.1. Study Site

This experiment was conducted in the greenhouse at the Jinnan Agricultural Science and Technology Demonstration Field (38°59′9.6″ N, 117°21′25.2″ E) at Nankai University, located in Tianjin City in North China. The area has a typical temperate monsoon climate, with an annual average air temperature of 13.4 °C, an annual average precipitation of 580.1 mm, and an annual average relative humidity of 59.0%. Recently, N deposition has increased rapidly in China, and the total amount of N deposition in Tianjin was up to 3.5 g N m$^{-2}$ year$^{-1}$ [22].

### 2.2. Sampling Preparation and Experiment Design

The two tree species, *Pinus tabuliformis* and *Fraxinus chinensis*, were selected to plant in the experiment, because (1) both of them are the dominant tree species in urban green spaces in North China; (2) *P. tabuliformis* is coniferous, which belongs to the gymnosperm, while *F. chinensis* is a broadleaved tree species of angiosperms. As the two common greening woody species, they have distinct growth strategies, functional traits, and responses to environmental gradients [23].

There were 36 healthy saplings chosen and planted in the greenhouse in February 2022. One-year-old bare-rooted seedlings of the two tree species (*P. tabuliformis* and *F. chinensis*) were transplanted into pots, which were provided by Dianzhuang Seeding Company, Jiangsu, China, and each seedling had a similar size (approximately 50 cm height). The cylindrical pots that were selected had a height of 40 cm and a base radius of 19 cm, and every two plastic pots were 20 cm apart. The soil was 0–20 cm-depth weathered soil (surface soil) from the artificial forests of *Pinus tabuliformis* and *Fraxinus chinensis* at Nankai University, which could represent the soil of an urban green space under natural conditions. After removing stones and other impurities, the soil was mixed homogeneously and three-quarters of each pot was filled with the soil, which weighed about 30 kg. The soil pH was $7.75 \pm 0.06$, the soil total C and N contents were $10.60 \pm 0.61$ mg g$^{-1}$ and $0.87 \pm 0.13$ mg g$^{-1}$, and the ammonium and nitrate N contents were $6.02 \pm 0.96$ μg g$^{-1}$ and $13.21 \pm 0.98$ μg g$^{-1}$, respectively. All pots were irrigated twice a week to keep the soil surface moist until the seedlings were adapted to the greenhouse conditions.

At the end of the acclimation period, a two-factorial experiment was conducted, including two tree species (*Pinus tabuliformis* and *Fraxinus chinensis*) and three N addition treatments (no N, low N ($3.5$ g N m$^{-2}$ year$^{-1}$), and high N addition ($10.5$ g N m$^{-2}$ year$^{-1}$)), with six replicates. The amount of N addition was based on the total amount of N deposition and its triple in Tianjin [4,22]. An aqueous solution of $NH_4NO_3$ salts was used for the different N addition gradients, added to the soil surface once a month. Meanwhile, the no N addition group received the same amount of water every time. The pots were watered twice a week and the amount of watering was selected to simulate the normal mean annual precipitation standard in Tianjin ($600$ mm year$^{-1}$). The mean temperature of the greenhouse was $24.37 \pm 0.06$ °C and the mean humidity was $64.67 \pm 0.18\%$ RH during the seedling growth.

### 2.3. Plant and Soil Sampling and Analysis

The saplings were cultivated from March to October 2022, which was the main growth period for the two tree species [21]. At the end of the experiment, the tree aboveground and belowground functional traits were measured based on the standard methods of Pérez-Harguindeguy et al. [17], which were universally validated as appropriate predictive indicators to reflect the plant resource acquisition, utilization, and storage strategies, including water, light, and nutrients (Table S1) [24]. Before harvest, plant height (PL) was measured as the vertical distance between the highest leaf of the tree in the natural state and the pot topsoil level. Then, five whole and expanded leaves of each plant were selected randomly to scan and measure the leaf area (LA) by ImageJ software (version 1.51j8; National Institutes of Health, Bethesda, MD, USA). Leaf thickness (LT) was calculated by measuring the overlapped leaves using a Vernier caliper. The fresh mass of the scanned leaves was also weighed and dried at 70 °C for 72 h to a constant weight and weighed. Specific leaf area (SLA) was calculated as the ratio of LA to leaf dry mass. Leaf dry matter content (LDMC) was calculated as the leaf dry mass divided by leaf fresh mass. Leaf tissue density (LTD) was calculated as the ratio of the leaf dry mass to its volume.

Subsequently, the plants were taken out of the pots to ensure the integrity of the roots, and then carefully washed with water and divided into the aboveground and belowground parts by cutting at the growing point. The fresh root samples were scanned and analyzed by the WinRHIZO root analysis system to obtain the total root length (RL), surface area of roots (RA), average root diameter (RD), root volume, and root tips. The root samples

were also dried and the following calculated: specific root length (SRL; RL/root dry mass), specific root area (SRA; RA/root dry mass), branching intensity (BRI; root tips/RL), and root tissue density (RTD; root dry mass/volume). All samples were also oven-dried to a constant weight at 70 °C for 72 h, weighed and the following calculated: the aboveground, belowground, and total biomass (AGB, BGB, Tbio) and the root–shoot ratio (R/S).

The rhizosphere soils were collected in the pots according to the shaking root method [25], and then air-dried and sieved through a 2 mm mesh to determine the effects of the different N addition and plant species on the soil nutrient properties. Similarly, dried plant samples were also ball-milled to fine powders to measure the leaf and root nutrient traits. The soil total carbon (STC) and total nitrogen (STN) content and tree leaf and root carbon and nitrogen content (LCC, RCC, LNC, and RNC) were measured using the Vario MAX C/N-Macro Elemental Analyzer and then the LCC/LNC and RCC/RNC were calculated. In addition, the soil ammonium (SNH) and nitrate (SNO) N contents were extracted in 2 mol/L KCl solution and determined by the Bran-Luebbe Flow Injection Analyzer. The soil available phosphorus (SAP) content was digested by a mixture of 0.03 mol/L $NH_4F$ and 0.025 mol/L HCl solution and determined by the Bran-Luebbe Flow Injection Analyzer.

### 2.4. Statistical Analysis

Before statistical analyses of variance (ANOVA), the Kolmogorov–Smirnov normality distribution test and the Bartlett homogeneity test of variances were applied. All data were normalized using log transformation before analysis when necessary.

Two-way ANOVAs were performed to test the effects of N addition, tree species, and their interaction on plant biomass, functional traits, and soil nutrient elements. An LSD multiple comparison test was employed to compare the means, and the significance level was $p < 0.05$. The statistical analyses were performed using SPSS 22.0. Simultaneously, to assess the relationship between the aboveground and belowground biomass, linear regressions were performed to calculate the slope and intercept. Pearson's correlation analysis was also used to test the connection between plant and soil traits. All of these analyses and figures were constructed using the package "psych", "car", and "vegan" in R version 4.1.1 [26].

To explore the underlying mechanisms of N addition on soil and plant nutrient elements and root–shoot ratio, structural equation modeling (SEM) analysis was used based on the regression analysis and previous experience. In this model, N addition was regarded as an exogenous variable, soil (STC, STN, and SAP) and tree (LNC, LCC, RNC, and RCC) nutrient traits were treated as endogenous variables, and R/S was considered as a response variable. All variables were standardized to satisfy the assumptions of normality and linearity. After removing non-significant pathways ($p > 0.05$) and collinear variables, Fisher's C test was used to evaluate the goodness of SEM and ensure its $p > 0.05$ with the lowest AIC value. SEM analyses were performed using the package "piecewiseSEM" in R.

## 3. Results

### 3.1. Aboveground, Belowground, and Total Biomass and Their Biomass Allocation

N addition and tree species significantly affected the plant aboveground, belowground, and total biomass. In addition, their interaction had a significant effect on the belowground biomass (Table S2). N addition significantly increased the total biomass of both *P. tabuliformis* and *F. chinensis*. Compared to the no N groups, low and high N addition enhanced the total biomass by 32.8% and 45.3% for *P. tabuliformis* and 33.1% and 47.4% for *F. chinensis*, respectively (Figure 1c). Specifically, N increased the AGB and BGB of *P. tabuliformis* and the BGB of *F. chinensis* (Figure 1a,b). Despite the AGB of *F. chinensis* performing an increasing trend with the supply of N, it was not significant and generally larger than the *P. tabuliformis*. Meanwhile, N addition significantly exacerbated the difference in BGB between *P. tabuliformis* and *F. chinensis*, leading to a root biomass of *F. chinensis* much larger than *P. tabuliformis*.

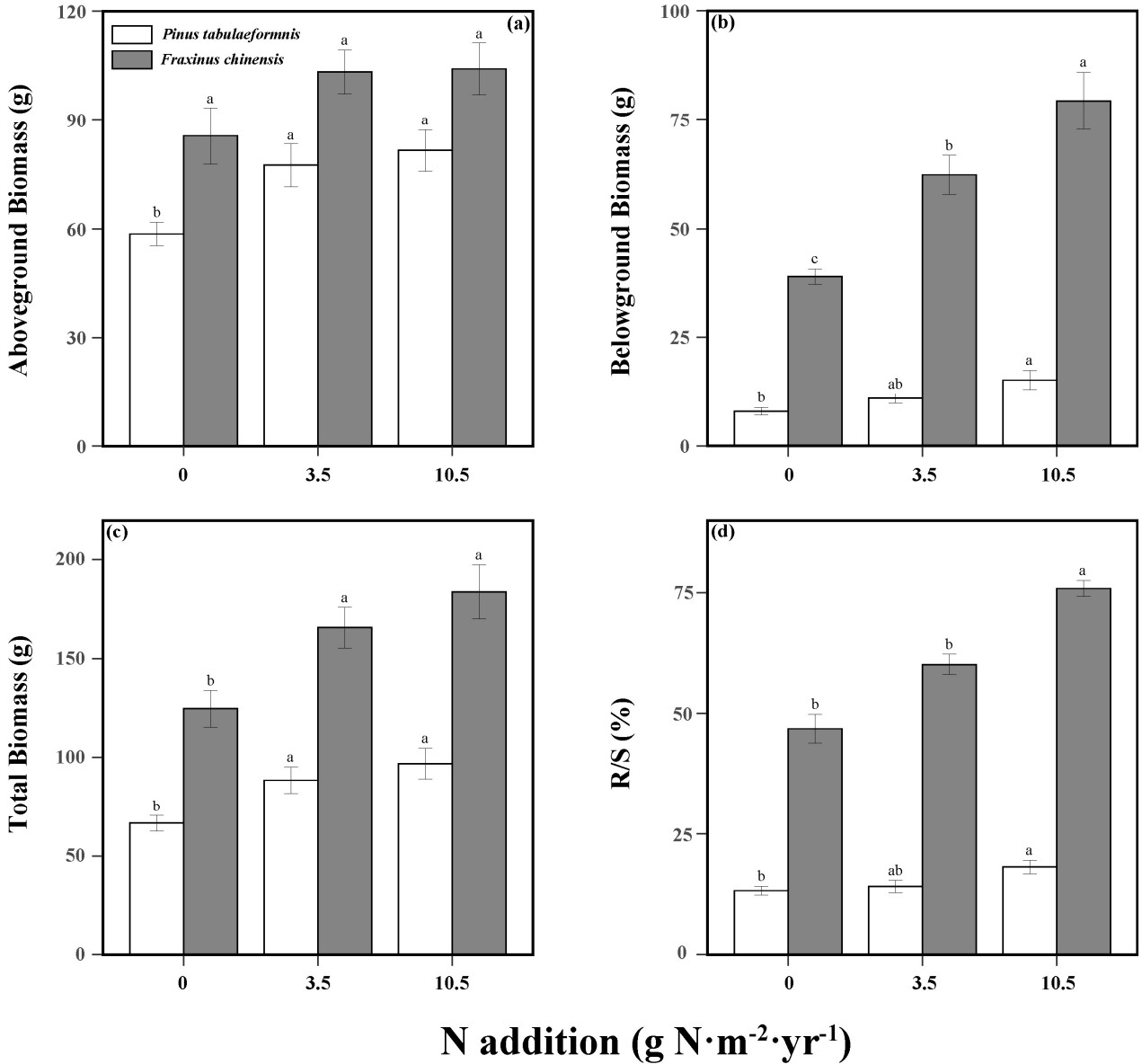

**Figure 1.** Aboveground biomass (**a**), belowground biomass (**b**), total biomass (**c**), and root−shoot ratio (R/S; (**d**)) of *Pinus tabuliformis* and *Fraxinus chinensis* under different N addition levels. Bars and error bars show the means ± SE. Different lowercase letters indicate significant differences among N treatments at the same time at a significance level of *p*-value < 0.05.

Furthermore, consecutive N addition and tree species had a significant interactive effect on plant biomass allocation (Table S2). The R/S of *F. chinensis* was significantly larger than *P. tabuliformis* and both of them were increased with N supply (Figure 1d). As for *P. tabuliformis*, N addition increased R/S by 6.7% under low N treatment and 36.7% under high N treatment relative to no N treatment, while the R/S of *F. chinensis* increased by 28.6% and 62.0%.

The results showed that there were positive relationships between the AGB and BGB of both the two target trees. The scaling slopes for the ratio of AGB to BGB of *P. tabuliformis* were 0.40, 0.41, and 0.44 in the no, low, and high N groups, and the correlation coefficients were 0.82, 0.12, and 0.64, respectively (Figure 2a). Despite there being no correlation in the low N treatment, it was obvious that the aboveground and belowground biomass allocation of *P. tabuliformis* showed an isometric relationship as N increased. As for *F. chinensis*, the scaling slopes for the ratio of AGB to BGB were 1.64, 0.68, and 0.79, and the correlation coefficients were 0.60, 0.74, and 0.93 under no, low, and high N treatment, respectively

(Figure 2b). These results showed that N addition significantly enhanced the belowground biomass allocation of *F. chinensis*.

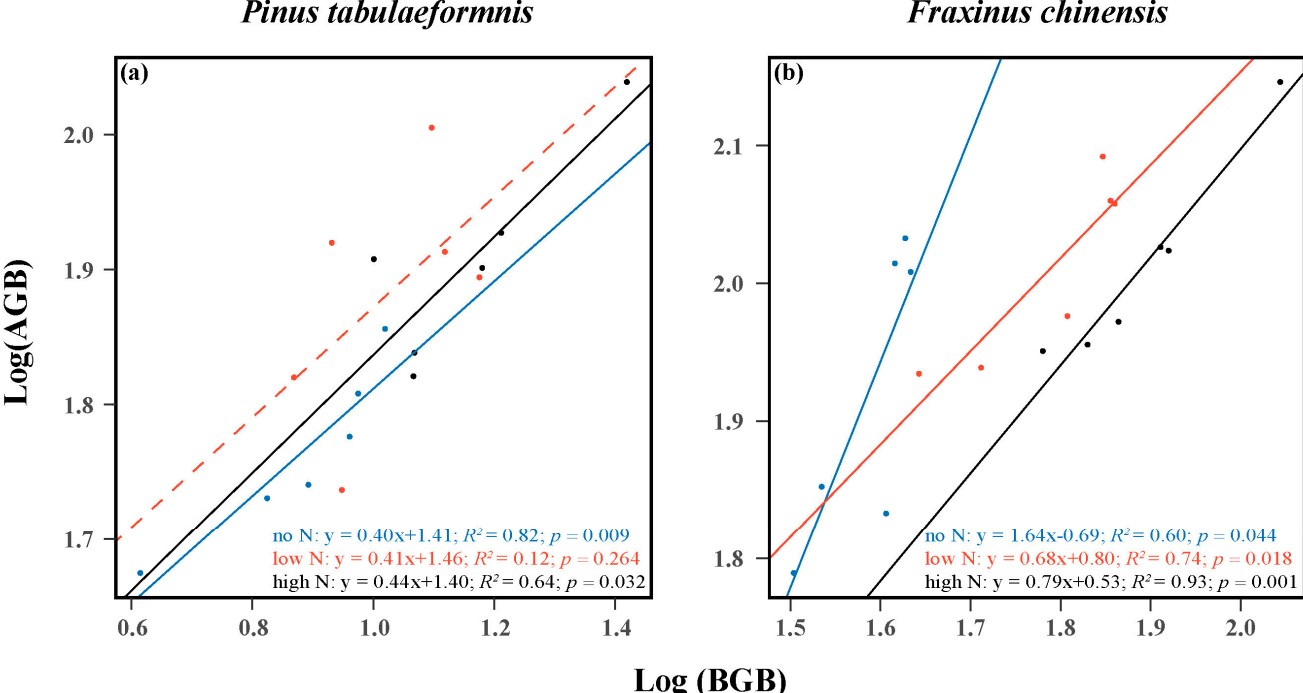

**Figure 2.** Relationships between the aboveground biomass (AGB) and belowground biomass (BGB) of *Pinus tabuliformis* (**a**) and *Fraxinus chinensis* (**b**). The $R^2$ (coefficient of determination), *p*-values, and equations are obtained from the linear regression analyses. Different color lines denote the allocation relationships under different N addition levels. Solid lines indicate significant confidence intervals ($p < 0.05$), while dotted lines indicate insignificant confidence intervals.

### 3.2. Aboveground and Belowground Functional Traits

N addition and tree species significantly affected the tree aboveground traits (Table S3). N increased the tree height, SLA, and LDMC of *P. tabuliformis*, while only high N promoted the LDMC and LTD of *F. chinensis*. There were no significant differences among different N treatments in the other aboveground traits of the two trees (Figure 3). The responses of belowground traits to N addition and tree species were significant, and the two factors had an interactive effect on the SRL, SRA, RD, and RTD (Table S4). N increased the SRL, RA, and SRA of *P. tabuliformis*, while reducing its RTD. As for *F. chinensis*, there were also positive relationships between N and RL, SRL, RA, and SRA, while there was a negative relationship between N and RD (Figure 4).

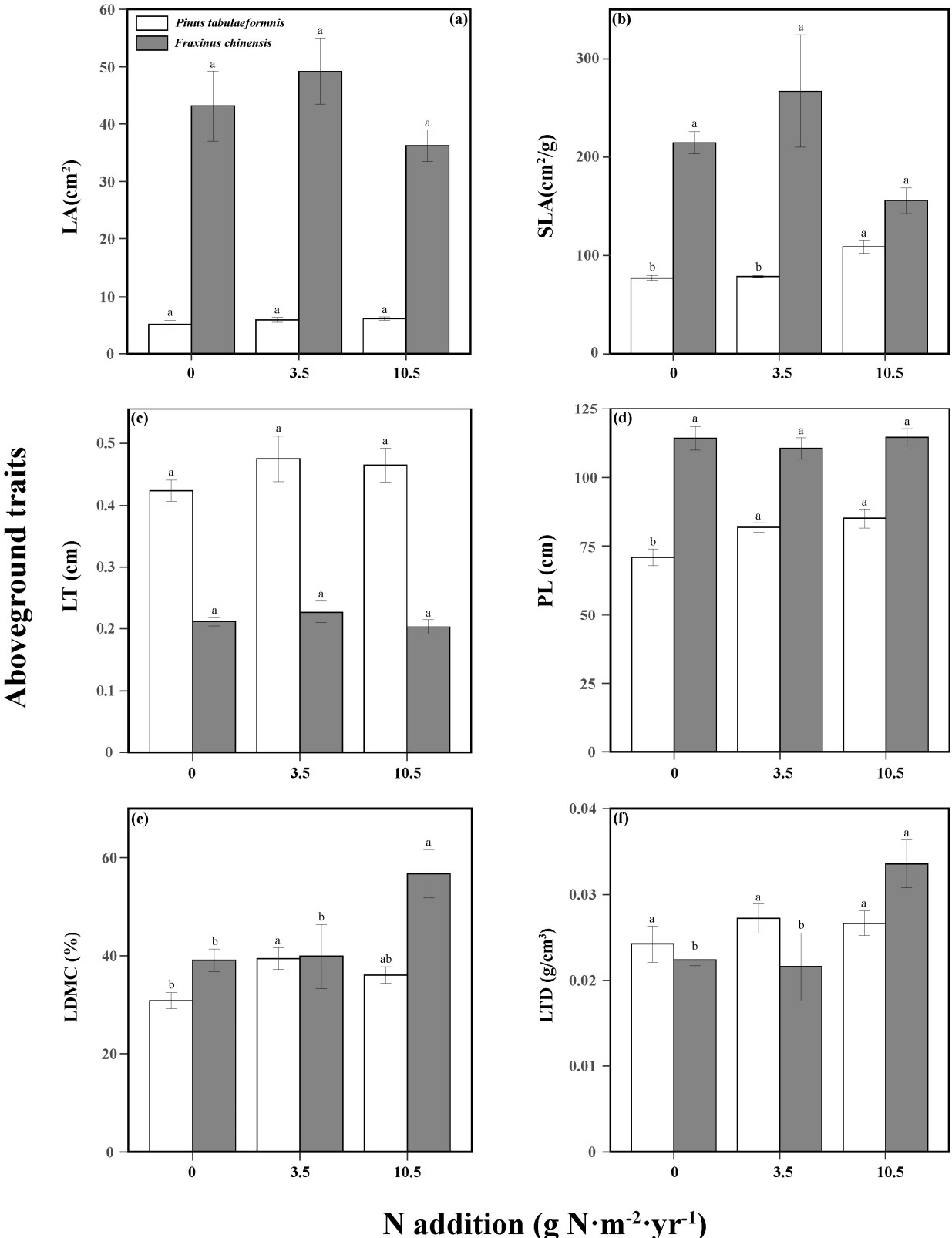

**Figure 3.** Aboveground traits including the leaf area (LA, (**a**)), specific leaf area (SLA, (**b**)), leaf thickness (LT, (**c**)), plant height (PL, (**d**)), leaf dry matter content (LDMC, (**e**)), and leaf tissue density (LTD, (**f**)) of *Pinus tabuliformis* and *Fraxinus chinensis* under different N addition levels. Bars and error bars show the means ± SE. Different lowercase letters indicate significant differences among N treatments at the same time at a significance level of *p*-value < 0.05.

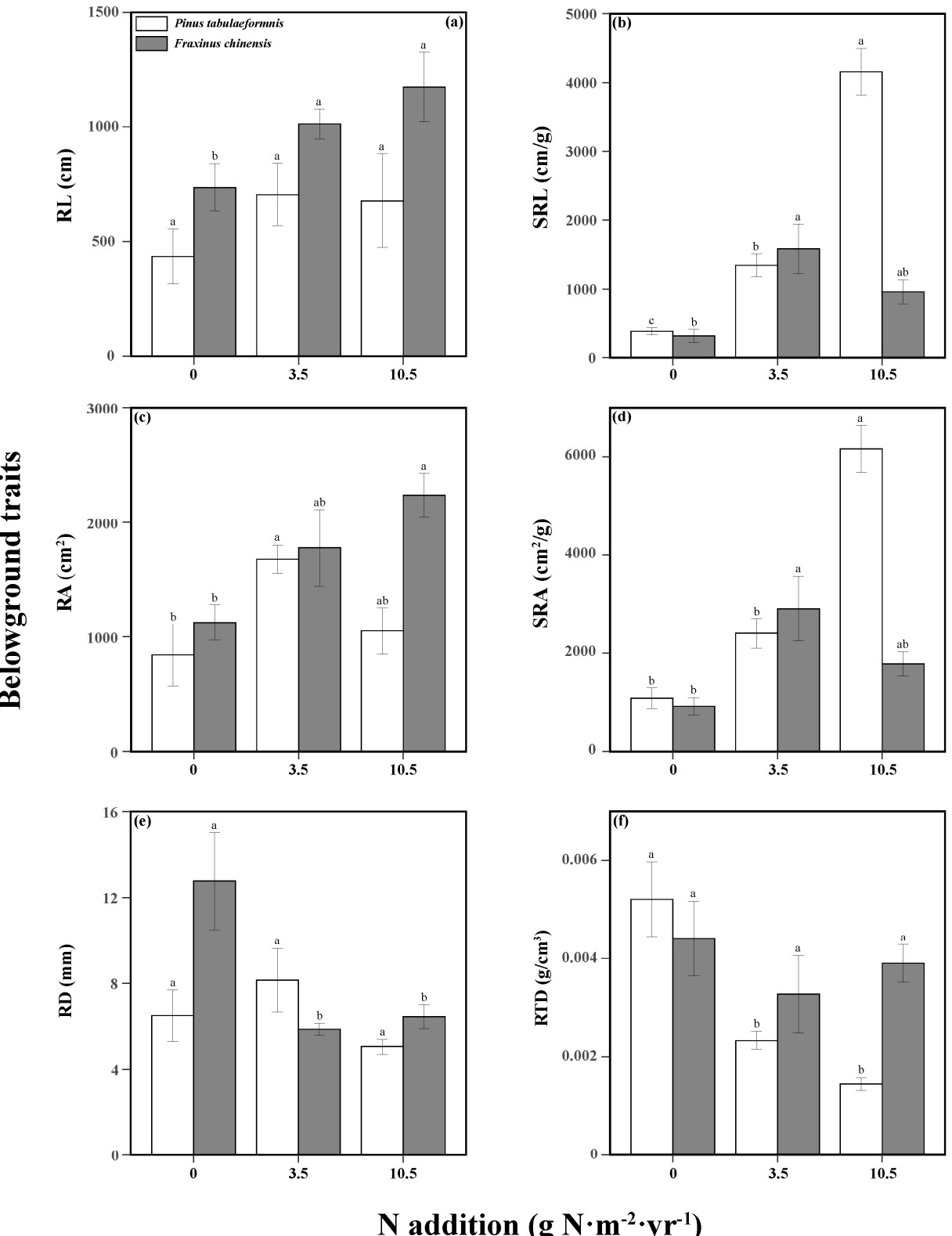

**Figure 4.** Belowground traits including the root length (RL, (**a**)), specific root length (SRL, (**b**)), root area (RA, (**c**)), specific root area (SRA, (**d**)), average root diameter (RD, (**e**)), and root tissue density (RTD, (**f**)) of *Pinus tabuliformis* and *Fraxinus chinensis* under different N addition levels. Bars and error bars show the means ± SE. Different lowercase letters indicate significant differences among N treatments at the same time at a significance level of *p*-value < 0.05.

In addition, a positive connection between the aboveground and belowground traits of *P. tabuliformis* was observed in response to N addition (Figure 5 and Figure S1a). Both tree height and SLA were significantly positively correlated with SRL and SRA. However, there was no obvious correlation between the aboveground and belowground traits of *F. chinensis* (Figures S1b and S2).

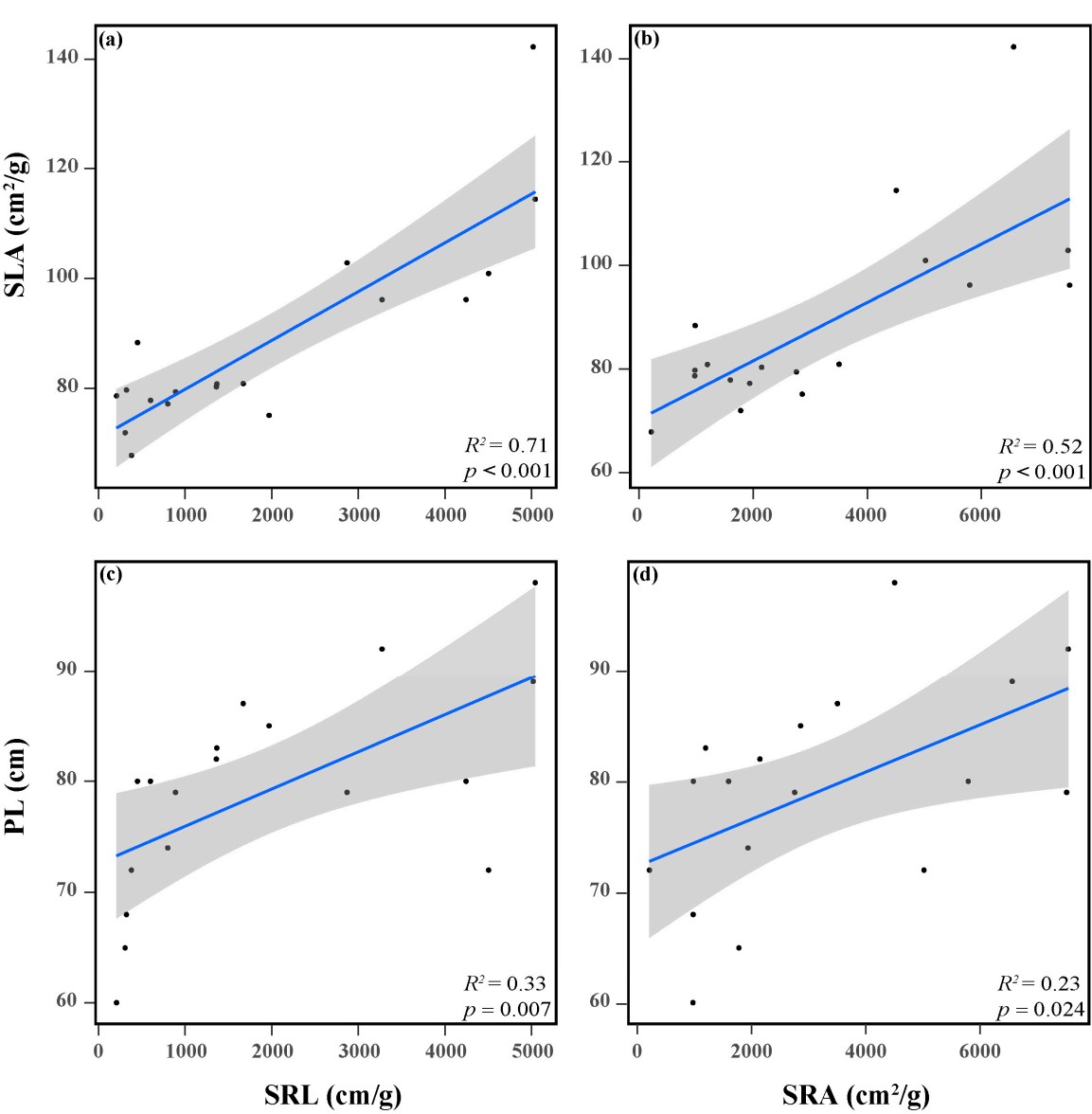

**Figure 5.** Relationships between SLA and SRL (**a**), SLA and SRA (**b**), PL and SRL (**c**), and PL and SRA (**d**) of *Pinus tabuliformis*. Trait acronyms: SLA = specific leaf area, PL = plant height, SRL = specific root length, SRA = specific root area. The $R^2$ (coefficient of determination), and *p*-values are obtained from the linear regression analyses. Shadow areas show a 95% confidence interval of the fit test. Solid lines indicate significant confidence intervals (*p* < 0.05), and the dots indicate the functional traits of the tree species.

### 3.3. Relationship between Tree Biomass and Functional Traits

The results of Pearson's correlation analysis indicated that there were significant associations between tree biomass and aboveground and belowground functional traits of *P. tabuliformis* (Figure S1a). PL and SLA were positively correlated with AGB, BGB, Tbio,

and R/S, while LDMC and LTD were positively correlated with AGB and Tbio. Likewise, SRL was positively correlated with AGB, BGB, Tbio, and R/S, and SRA was positively correlated with BGB, Tbio, and R/S. However, RTD was negatively correlated with AGB, BGB, and Tbio. As for *F. chinensis*, only the belowground traits were significantly correlated with biomass (Figure S1b). RL and RA were positively related to BGB, Tbio, and R/S, while RD was negatively connected with AGB, BGB, and Tbio.

### 3.4. Nutrient Elements Traits of Trees and Soil

Tree nutrient traits were significantly affected by N addition, tree species, and their interaction (Table S5). N increased the RCC and RNC of *P. tabuliformis*, while decreasing the LCC/LNC and RCC/RNC. Similarly, the LNC and RNC of *F. chinensis* were increased and the LCC/LNC and RCC/RNC were decreased as N increased (Figure 6). There was no significant effect of N on LCC and RCC. As for soil properties, N increased the STN, SAP, SNH, and SNO in the rhizosphere soil of both tree species, but did not change the STC (Table 1).

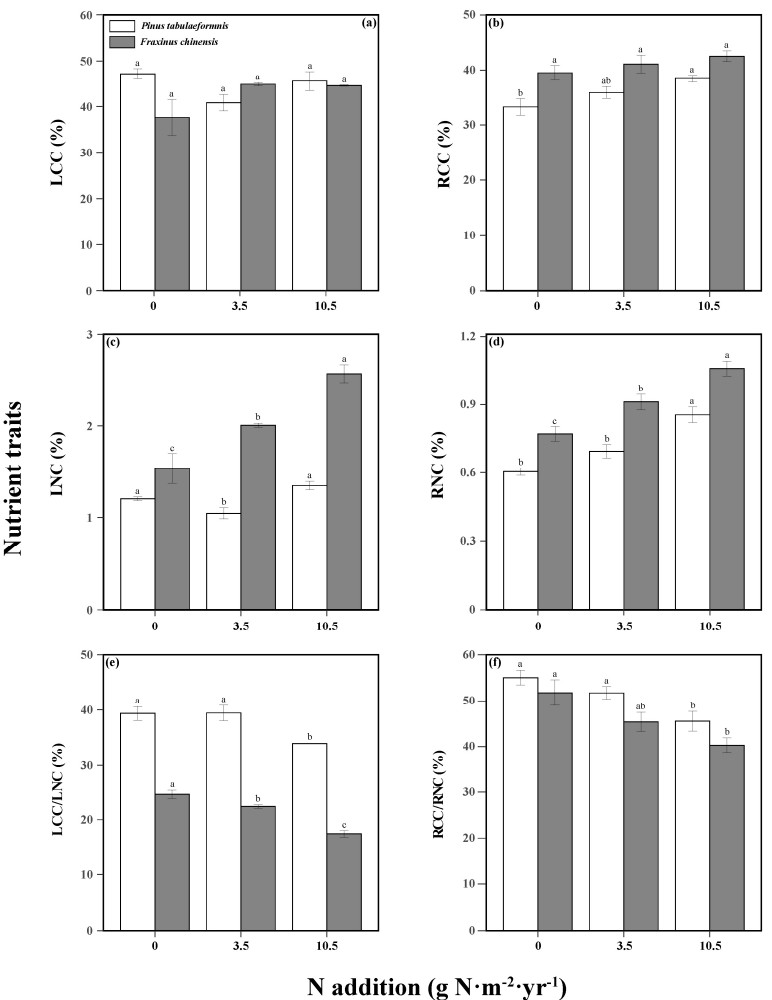

**Figure 6.** Nutrient traits including the leaf carbon content (LCC, (**a**)), leaf nitrogen content (LNC, (**b**)), root carbon content (RCC, (**c**)), root nitrogen content (RNC, (**d**)), LCC/LNC (**e**), and RCC/RNC (**f**) of *Pinus tabuliformis* and *Fraxinus chinensis* under different N addition levels. Bars and error bars show the means ± SE. Different lowercase letters indicate significant differences among N treatments at the same time at a significance level of *p*-value < 0.05.

**Table 1.** Soil nutrients include the total carbon (STC), total nitrogen (STN), ammonium nitrogen (SNH), nitrate nitrogen (SNO), and available phosphorus (SAP) concentrations under different N treatments and tree species. Data are the means ± SE (*n* = 6). Different lowercase letters indicate significant differences among N treatments at the same time at a significance level of *p*-value < 0.05.

| Treatment | STC (mg/g) | STN (mg/g) | SNH (mg/kg) | SNO (mg/kg) | SAP (mg/kg) |
|---|---|---|---|---|---|
| *Pinus tabuliformis* | | | | | |
| No N | 13.63 ± 1.93 | 1.12 ± 0.04 b | 6.52 ± 0.70 c | 14.92 ± 1.06 b | 7.84 ± 0.41 b |
| Low N | 9.68 ± 0.88 | 1.18 ± 0.06 b | 7.88 ± 0.26 b | 15.68 ± 0.54 b | 11.01 ± 1.39 a |
| High N | 13.47 ± 0.18 | 1.38 ± 0.07 a | 9.73 ± 2.36 a | 16.84 ± 0.11 a | 12.12 ± 0.69 a |
| *Fraxinus chinensis* | | | | | |
| No N | 8.07 ± 0.17 | 0.67 ± 0.04 b | 2.43 ± 0.15 b | 12.18 ± 1.63 b | 8.41 ± 0.27 b |
| Low N | 8.33 ± 0.21 | 0.81 ± 0.02 a | 4.75 ± 1.08 a | 22.96 ± 1.45 a | 8.48 ± 0.60 b |
| High N | 8.20 ± 0.03 | 0.82 ± 0.04 a | 4.20 ± 0.90 a | 20.52 ± 0.15 a | 10.06 ± 0.42 a |

The first structural equation model fitted the variance best and explained 94% of the variance in the R/S of *P. tabuliformis* (Figure 7a). The SEM showed that N addition had direct inhibiting and indirect promoting effects on R/S (Table 2). Specifically, N addition indirectly increased R/S through positive effects on RNC and STN, and the sum of these effects was greater than the direct inhibiting effect. Consequently, increased N addition indirectly positively affected R/S by regulating soil and root N concentration. In addition, N could also increase RNC via an indirectly altering STN, while decreasing LNC through indirectly increasing SAP.

The SEM2 explained 96% of the variance in R/S of *F. chinensis* (Figure 7b). N addition was positive with soil traits (STN, SAP) and tree nutrient elements (LNC, RNC). The result showed that N increased R/S through direct and indirect effects (Table 2). Specifically, N addition could promote root N concentration via a direct effect or indirectly increasing soil N and P availability, and finally indirectly increasing R/S.

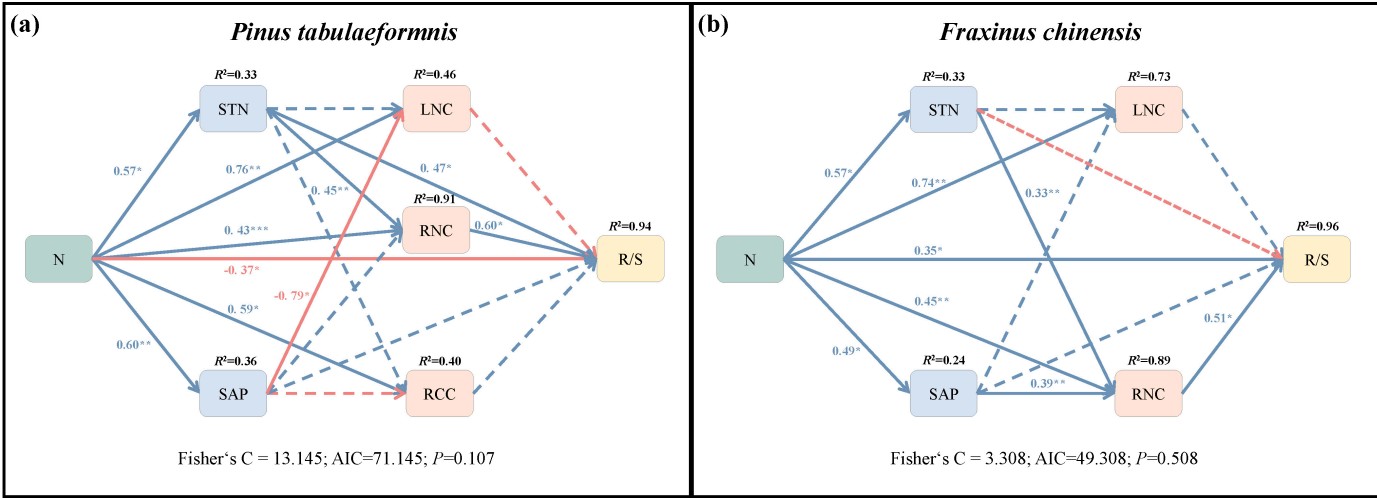

**Figure 7.** Structural equation model (SEM) showing the direct and indirect effects of N addition on the root−shoot ratio (R/S) and soil N, P content (SNC, SAP) and root C, N content (RCC, RNC) and leaf N content (LNC) of Pinus tabuliformis (**a**) and Fraxinus chinensis (**b**). The single-headed arrows represent paths in this conceptual model. The blue and red arrows separately indicate positive and negative pathways. The solid and dotted lines separately indicate significant (*p* < 0.05) and insignificant pathways. Numbers at arrows are standardized path coefficients and the proportion of variance explained ($R^2$) appears at the top of response variable in the model. *, **, and *** indicate statistically significant paths at 0.01 < *p* < 0.05, 0.001 < *p* < 0.01, and *p* < 0.001, respectively.

**Table 2.** Direct, indirect, and total effects on the root−shoot ratio (R/S) on standardized values of statistically significant SEM paths ($p < 0.05$). The direction of the relationship is indicated by + (positive relationship) or − (negative relationship). NS indicates no significant relationships.

| Predictor | Pathway to R/S | Effect |
|---|---|---|
| *Pinus tabuliformis* | | |
| N | Direct | −0.37 |
| | Indirect | 0.68 |
| | Total | 0.31 |
| STN | Direct | 0.47 |
| | Indirect | 0.27 |
| | Total | 0.74 |
| RNC | Direct | 0.60 |
| | Indirect | NS |
| | Total | 0.60 |
| *Fraxinus chinensis* | | |
| N | Direct | 0.35 |
| | Indirect | 0.42 |
| | Total | 0.77 |
| RNC | Direct | 0.51 |
| | Indirect | NS |
| | Total | 0.51 |

## 4. Discussion

To explore whether N deposition could affect plant biomass allocation by changing functional traits, two dominant urban greening tree species were selected to carry out a short-term N deposition simulation experiment. With the increase in N application, different types of trees exhibited different growth strategies and chose different biomass allocation patterns. Meanwhile, functional traits were the determinants of tree species' responses to N addition. The increased N availability changed the soil nutrient properties and plant aboveground and belowground traits, and was reflected in the biomass allocation. Trees could adapt to the high N deposition environments of urban green spaces by modifying root nutrient traits, which finally changed the root structures and biomass allocation strategies.

### 4.1. Response of Tree Biomass Allocation to Nitrogen Addition

Many studies have confirmed that N has a positive effect on plant growth and biomass production, both in the aboveground and belowground parts [10,27]. Our results demonstrated that N addition increased the biomass of both *P. tabuliformis* and *F. chinensis*, but the biomass allocation strategies of the two species were different. N addition promoted both AGB and BGB of *P. tabuliformis*, and there existed isometric relationships between AGB and BGB. This indicated that the aboveground and belowground growth patterns of *P. tabuliformis* were consistent in the high N deposition environment, similar to in the work of Yang et al. [11]. However, it did not substantiate our hypothesis that the tree biomass allocation strategy tended to optimal partitioning. Compared with hardwoods, the root biomass allocation of conifers was relatively low, with lower root production and turnover [23,28]. *P. tabuliformis* could adapt to the poorer soil fertility environment (higher soil carbon/nitrogen ratio) mainly through prioritizing the allocation of resources to leaves and stems, such as photosynthetic organs, rather than increasing root nutrient uptake [29]. Moreover, relatively short-term environmental fluctuations might not be sufficient to reflect plant biomass allocation, so that *P. tabuliformis* performed an isometric growth strategy to meet the biomass needs of different plant organs [14].

However, as for *F. chinensis*, exogenous N supplies only increased the belowground part for *F. chinensis*, and thus increased the proportion of root biomass allocation. Generally, due to the higher root variability of hardwoods, their saplings allocate more woody biomass to roots than conifer species as the soil resources change [30]. In addition, N is a limited

element in boreal and temperate forests, and the native soil N content ($0.87 \pm 0.13$ mg g$^{-1}$) in this study was at a poor nutrition level according to the Second National Soil Survey of China [31]. This meant *F. chinensis* would invest more resources in root biomass, leading it to facilitate root extension and improve the resource-use efficiency in the soil [32]. N addition improved the soil N availability and promoted root growth to gain more nutrient resources, thereby leading the tree species to adapt to the resource-poor environment [33]. On the contrary, the aboveground organs might be less important and did not change significantly as N increased. Therefore, the response of *F. chinensis* to N addition followed the ratio-based optimal partitioning theory that the biomass allocation pattern was induced by environmental change and also proved the first hypothesis [9].

### 4.2. Response of Tree Functional Traits to Nitrogen Addition

N affected both the aboveground and belowground functional traits of *P. tabuliformis*. Under N fertilization, the tree height and SLA of *P. tabuliformis* tended to be higher, indicating that the coniferous tree could improve the capacity to capture light energy and promote leaf photosynthetic efficiency [34]. Thus, *P. tabuliformis* might adapt to the high N deposition environment and increase the aboveground biomass mainly through increasing the aboveground acquisitive traits. Meanwhile, the SRL and SRA of *P. tabuliformis* were increased and RTD decreased as N increased. SRL and SRA were acquisitive traits, which generally indicated the efficiency of root resource foraging and were related to root resource acquisition capacity, while RTD tended to be conservative and was negatively correlated with root growth [35,36]. N addition changed the root architecture and shifted the resource strategy to the fast pattern, thus improving the root resource acquisition efficiency and biomass production. In addition, the results also showed that the variations in aboveground and belowground functional traits of *P. tabuliformis* were consistent, similar to the tree biomass allocation pattern. The responses of the aboveground and belowground parts of *P. tabuliformis* to N deposition had synergetic effects, and showed the same growth and resource acquisition strategy [37].

However, N only promoted the root acquisitive traits of *F. chinensis*, but was positively correlated with leaf conservative traits, which also coincident with the response of the biomass allocation pattern. Compared to conifers, broadleaf saplings had a lower root morphological similarity (monopodial growth) and more apparent variability when adapting to environmental changes [38]. Therefore, the root traits of *F. chinensis* were more sensitive than those of *P. tabuliformis* as soil N availability increased. Moreover, according to optimal partitioning theory, *F. chinensis* would invest more energy into the root system in the poor soil, which could modify the root architecture and increase the root biomass allocation [39]. N addition increased the surface length and area of roots and improved the root nutrient acquisition capacity of *F. chinensis*, promoting better growth under nutrient-poor conditions [6,14]. In contrast, the leaves showed a conservative strategy in response to N addition. As few resources were put into the construction of aboveground organs, plants would only increase the nutrient storage of leaves to support their growth [40]. Overall, tree aboveground and belowground traits showed the opposite resource acquisition strategies as N inputs, which was in line with that of Asefa et al. [41] and confirmed the second hypothesis.

### 4.3. Response of Soil and Tree Nutrient Elements to Nitrogen Addition

The finding that N supply significantly increased the soil N element availability (STN, SNH, SNO, SAP) of both *P. tabuliformis* and *F. chinensis*, and thus improved the LNC and RNC of the target trees, is consistent with our previous study [15]. With the improvement of soil N availability, plants with a higher LNC tended to possess greater capacities for foliar N uptake and storage, which promoted leaf photosynthesis and tree biomass accumulation [34,42]. Meanwhile, RNC was considered a vital predictor of root nutrient and respiration, and roots with high N contents could increase the metabolic activity and root nutrient uptake rates [18]. In addition, N addition decreased the LCC/LNC

and RCC/RNC, which changed the original element balance and thereby accelerated the cycle of C and N in the trees [15,43]. It is reported that evergreen and needleleaf species performed higher C/N to guarantee survival in a N limiting environment and at the slower position of the plant economic spectrum. However, they would transform towards low C/N ratios to synthesize protein faster and improve competitiveness as more N became available [39,43], which could also be appropriate for the results of this study.

The tree root–shoot ratio is a direct embodiment of the biomass allocation strategy and growth pattern, sensitive to environmental changes [10]. This study indicated that despite directly inhibiting the R/S of *P. tabuliformis*, N addition could indirectly increase it through the multiple direct and indirect positive effects of STN and RNC. The tree species could increase root biomass allocation mainly through absorbing more resources from soil [33], and soil N availability and root N uptake played important roles in the root growth of *P. tabuliformis*. As for *F. chinensis*, on the one hand, N could indirectly increase R/S through its positive effects on STN and RNC, consistent with *P. tabuliformis*. On the other hand, N inputs also improved RNC by increasing SAP, which might emphasize the responses of soil P to N addition [44]. N addition improved soil P availability and accelerated P cycling rates, which could increase the N demand of plants and alleviate the nutrient elements imbalance caused by excessive N uptake [45,46]. In short, N deposition changed the biomass allocation mainly through improving the RNC of the two-target conifer and broadleaf tree species, rather than LNC. Compared to aboveground parts, root functional traits were directly correlated to soil nutrient elements, and reflected in the root–shoot ratio [15]. Plants could modify the root structure to promote the uptake of nutrient concentration, so as to better adapt to the poor soil nutrient environment [12,14]. These findings support the last hypothesis and reveal the mechanism of N addition in tree biomass allocation by altering root nutrient traits.

*4.4. Limitations and Future Directions*

Based on the experiments, this study demonstrates how N addition affects biomass allocation and functional traits of two urban greening dominant tree species in North China. N deposition could influence the plant resource acquisition strategy and growth pattern, which differed between the two tree species. However, since the experiment was conducted in a greenhouse and the 1-year-old seedlings were treated with N addition for only seven months, our study may not consider the seasonal fluctuations in the natural environment and fully accurately reflect the responses of mature trees to long-term N deposition. Therefore, it is important to choose mature conifers and broadleaved tree species to further explore whether the results can be extended to a longer time scale in the future [4]. Furthermore, plant anatomical traits also exhibited a certain degree of plasticity in response to environmental changes, which were directly correlated with plant biomass allocation [47,48]. The underlying mechanisms whereby N addition affected the tree growth strategies through modifying the anatomical traits (i.e., xylem, leaf, or root tissue) remain to be explored. Further experiments should explore the trade-off between the economic and anatomical traits of different tree species and how it could be reflected in biomass allocation under different N addition treatments.

**5. Conclusions**

N addition affected the biomass allocation and functional traits between two tree species dominating in urban green space. It increased both the aboveground and below-ground biomass of *P. tabuliformis*, and the relationship between them was isometric, which was also reflected in the increase in tree acquisitive traits. However, only root biomass and fast root traits of *F. chinensis* significantly increased with N addition, while aboveground traits tended to be conservative due to the reduced resource inputs. Furthermore, N inputs increased the uptake of root N content by improving the soil N and P availability, and thus promoted root biomass allocation in the limited environment. Therefore, these findings could be better explained by the optimal partitioning theory that plants would invest

more resources in roots to adapt to the resource-poor areas. Our study provides a novel perspective into the mechanisms of N deposition in the urban greening tree functional traits and biomass allocation and puts forward a new tree growth strategy in response to global climate changes in silvicultural practice.

**Supplementary Materials:** The following supporting information can be downloaded at: https://www.mdpi.com/article/10.3390/f15010199/s1, Figure S1: Correlation matrix of plant functional traits and biomass of *Pinus tabuliformis* (a) and *Fraxinus chinensis* (b). Figure S2: Relationships between SLA and SRL (a), SLA and SRA (b), PL and SRL (c), and PL and SRA (d) of *Fraxinus chinensis*. Table S1: List of plant functional traits selected. Table S2: Effects of N addition (no, low, and high N) and tree species (*Pinus tabuliformis* and *Fraxinus chinensis*) on plant biomass (aboveground, belowground, and total biomass) and root−shoot ratio. Table S3: Effects of N addition (no, low, and high N) and tree species (*Pinus tabuliformis* and *Fraxinus chinensis*) on aboveground traits including the leaf area (LA), specific leaf area (SLA), leaf thickness (LT), plant height (PL), leaf dry matter content (LDMC), and leaf tissue density (LTD) of *Pinus tabuliformis* and *Fraxinus chinensis*. Table S4: Effects of N addition (no, low, and high N) and tree species (*Pinus tabuliformis* and *Fraxinus chinensis*) on belowground traits including the root length (RL), specific root length (SRL), root area (RA), specific root area (SRA), average root diameter (RD), root tissue density (RTD) and branching intensity (BRI) of *Pinus tabuliformis* and *Fraxinus chinensis*. Table S5: Effects of N addition (no, low, and high N) and tree species (*Pinus tabuliformis* and *Fraxinus chinensis*) on nutrient traits including the leaf carbon content (LCC), leaf nitrogen content (LNC), root carbon content (RCC), root nitrogen content (RNC), LCC/LNC, and RCC/RNC of *Pinus tabuliformis* and *Fraxinus chinensis*.

**Author Contributions:** Conceptualization, Q.Z., J.Z. and H.L.; methodology, Q.Z., J.Z., J.L., M.L. and H.L.; software, Q.Z.; validation, Q.Z., J.Z. and H.L.; formal analysis, Q.Z., J.L. and M.L.; investigation, Q.Z., J.L., M.L. and S.H.; resources, Q.Z., J.Z. and H.L.; data curation, Q.Z., J.Z. and S.H.; writing—original draft, Q.Z.; writing—review and editing, Q.Z., J.Z. and H.L.; visualization, J.L.; supervision, H.L.; project administration, H.L.; funding acquisition, H.L. All authors have read and agreed to the published version of the manuscript.

**Funding:** This research was funded by the National Natural Science Foundation of China, grant number 32171853.

**Data Availability Statement:** The data are available from the corresponding author on reasonable request.

**Conflicts of Interest:** The authors declare no conflicts of interest.

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
