# Peer review of "Urban Dominant Trees Followed the Optimal Partitioning Theory and Increased Root Biomass Allocation and Nutrient Uptake under Elevated Nitrogen Deposition"

_forests, doi:10.3390/f15010199_

Round 1
Reviewer 1 Report
Comments and Suggestions for Authors
This manuscript investigates the influence of nitrogen (N) availability in urban soil on biomass partitioning in two distinct tree species. The study offers a unique perspective on how varying N concentrations can affect trees with different characteristics in an urban environment. While the manuscript's quality is generally acceptable, some issues need addressing, and certain sections could be improved in terms of clarity. Here are some comments for the manuscript:
1. The use of third-person point of view is recommended throughout the article for consistency.
Introduction:
2. Consider incorporating previous studies on urban trees and biomass partitioning in response to changes in nutrient supply to provide a comprehensive background for the study. Although well-written, the introduction appears a bit long. By shortening it while retaining essential information will improve its clarity.
Materials and Methods:
3. Ln 99, did the study period from May to October 2022 encompass all seasonal changes in the region? If it is so, does the seasonal change affect the partitioning?
4. Provide information on soil nutrient content before N supplementation to establish a baseline understanding.
5. Elaborate on the rationale behind selecting two different tree species in terms of taxonomy, discussing potential differences in their responses to soil N availability.
6. Clarify if the soil source (greenhouse) is representative of the actual urban soil where these trees are planted. Urban trees are often exposed to varied soil conditions, such as sandy or ex-constructed soil with a high load of coarse particles.
Results:
7. Include units on the X axes of each graph for clarity.
Discussion:
8. Explore and discuss the physiological mechanisms within the trees that affect the responses to varying N concentrations in the soil.
9. Consider including simple observations of plant tissue, in addition to weight measurements. This information could provide insights into whether changes in biomass and R/S biomass partitioning correlate with plant anatomy.
Limitations and Future Directions:
10. Add a section discussing the limitations of the study and suggesting potential future directions. This will enhance the manuscript's significance by acknowledging constraints and proposing avenues for further research.
By addressing these suggestions, the manuscript can be refined for improved clarity, consistency, and depth of analysis.
Author Response
Independent Review Report, Reviewer 1
This manuscript investigates the influence of nitrogen (N) availability in urban soil on biomass partitioning in two distinct tree species. The study offers a unique perspective on how varying N concentrations can affect trees with different characteristics in an urban environment. While the manuscript's quality is generally acceptable, some issues need addressing, and certain sections could be improved in terms of clarity. Here are some comments for the manuscript:
- The use of third-person point of view is recommended throughout the article for consistency.
Response: Thanks for the reviewer’s suggestion.
The suggestion you put forward is very important for us. We have revised the manuscript and used the third-person point of view to ensure the consistency and objectivity of the article (See the revised manuscript).
Introduction:
- Consider incorporating previous studies on urban trees and biomass partitioning in response to changes in nutrient supply to provide a comprehensive background for the study. Although well-written, the introduction appears a bit long. By shortening it while retaining essential information will improve its clarity.
Response: Thanks for the reviewer’s suggestion.
Thank you very much for your approval of the Introduction to our manuscript. We have reduced the content of the Introduction according to your suggestion to improve its clarity. The Introduction of the original manuscript was consistent of five paragraphs and 823 words, while now it was shortened to four paragraphs and 718 words (See the Introduction in the revised manuscript).
Materials and Methods:
- Ln 99, did the study period from May to October 2022 encompass all seasonal changes in the region? If it is so, does the seasonal change affect the partitioning?
Response: Thanks for the reviewer’s suggestion.
The question you raised about the seasonal change is very important, and we also took this question into consideration at that time. First, I would like to express my apologies to you. Our experiment period is actually seven months, from March to October. We mistakenly wrote May instead of March, which has now been corrected in the manuscript.
In our study, the experiment period from March to October 2022 was the mainly growth period for the two tree species, when trees put a large amount of biomass to the growth of stems and leaves for photosynthesis and nutrient uptake [2,3]. We also have added it in the M&M (See L130-131 in the revised manuscript). As you mentioned, seasonal change could change the tree growth and biomass allocation, which can’t be ignored [1]. However, this study mainly focuses on the effects of N addition on biomass allocation and functional traits of tree species, so that we conducted a simulation experiment in a greenhouse. Compared to the natural ecosystem, the temperature and humidity of the greenhouse were always at a relatively high and stable level during the experiment period, where tree species are less affected by seasonal fluctuations [4]. Therefore, we only conducted the N addition experiment at the growth period of tree species, while not involved with seasonal change. However, your advice is very valuable to us, and we have included it in the Limitations and Future Directions of our paper (See L429-430 in the revised manuscript).
- Kimura, K.; Yasutake, D.; Koikawa, K.; Kitano, M. Spatiotemporal variability of leaf photosynthesis and its linkage with microclimates across an environment-controlled greenhouse. Biosyst. Eng. 2020, 195, 97-115.
- Liu, H.; Yin, C.; Hu, X.; Tanny, J.; Tang, X. Microclimate Characteristics and Evapotranspiration Estimates of Cucumber Plants in a Newly Developed Sunken Solar Greenhouse. Water 2020, 12, 2275.
- Hitsuma, G.; Han, Q.; Chiba, Y. Photosynthesis and growth of Thujopsis dolabrata var. hondai seedlings in the understory of trees with various phenologies. J. Forest Res. 2012, 17, 156-163.
- Yang, L.; Liu, H.; Tang, X.; Li, L. Tomato Evapotranspiration, Crop Coefficient and Irrigation Water Use Efficiency in the Winter Period in a Sunken Chinese Solar Greenhouse. Water 2022, 14, 2410.
- Provide information on soil nutrient content before N supplementation to establish a baseline understanding.
Response: Thanks for the reviewer’s suggestion.
As you mentioned, we have provided the information about soil nutrient content before N supplementation (See the L113-117 in the revised manuscript). The original soil pH was 7.75 ± 0.06, soil total C and N contents were 10.60 ± 0.61 mg g-1 and 0.87 ± 0.13 mg g-1, and ammonium and nitrate N contents were 6.02 ± 0.96 μg g-1 and 13.21 ± 0.98 μg g-1, respectively. While the soil nutrients under different N treatments (no, low, and high N supplementation) were shown in Table 1 in the revised manuscript. This can establish a baseline to better understand the results of our study and make it more reasonable.
- Elaborate on the rationale behind selecting two different tree species in terms of taxonomy, discussing potential differences in their responses to soil N availability.
Response: Thanks for the reviewer’s suggestion.
The two tree species, Pinus tabuliformis and Fraxinus chinensis, were chosen to plant in our experiment, because (1) both of them are the dominated tree species in urban green space in North China, which play the important role in urban green space. (2) Pine is coniferous, while ash is deciduous. There are morphological and anatomical differences in their leaves and seeds (See L75-77 in the Introduction of the revised manuscript). (3) Pinus tabuliformis was the gymnosperm, while Fraxinus chinensis belongs to angiosperm, which have distinct growth and development strategies in response to climate change [1,2]. For example, a global meta-analysis has revealed that angiosperms and gymnosperms differ in many basic functional traits as the two major extant clades of woody species [3,4]. Therefore, it is necessary to explore the response of different coniferous and broadleaved tree species to N deposition in urban green space, so as to better understand the tree growth patterns and aboveground and belowground resource acquisition strategies to adapt to high soil N availability. Meanwhile, according to your suggestion, we added the reason why we selected the two tree species in the M & M, and we explained it in terms of taxonomy, too (See L98-103 in the revised manuscript).
Few previous studies have focused on the response of urban greening tree species to N deposition. In our Discussion, we have explored the potential differences between Pinus tabuliformis and Fraxinus chinensis in response to soil N availability. Due to the difference root morphological traits [2,5], broadleaf saplings had a larger root variability when adapting to the environmental changes, which could put more resource into the root to modify the root architecture and acquire more nutrients, so that thus increased the root biomass. On the contrary, the aboveground organs might be gained fewer resource input and didn’t change significantly as N increased (See L343-353 and L375-388 in the revised manuscript). While P. tabuliformis had a higher root morphological similarity (monopodial growth), and the root biomass allocation was relatively low, with lower root production and turnover. So, P. tabuliformis adapted to the high soil N availability environment mainly through prioritizing the allocation resources to leaves and stems, such as photosynthetic organs, rather than only increasing root nutrient uptake. In addition, the responses of aboveground and belowground parts of P. tabuliformis to soil N availability had synergetic effects, and showed the same growth and resource acquisition strategy (See L334-341 and L358-372 in the revised manuscript). Overall, your suggestions are very valuable to us, and we have added them in the Discussion (See Discussion in the revised manuscript).
- Ma, Z.; Sandel, B.; Svenning, J. Phylogenetic assemblage structure of North American trees is more strongly shaped by glacial–interglacial climate variability in gymnosperms than in angiosperms. Ecol. Evol. 2016, 6, 3092-3106.
- Wang, C.; McCormack, M.L.; Guo, D.; Li, J. Global meta‐analysis reveals different patterns of root tip adjustments by angi-osperm and gymnosperm trees in response to environmental gradients. J. Biogeogr. 2018, 46, 123-133.
- Pregitzer, K.S.; DeForest, J.L.; Burton, A.J.; Allen, M.F.; Ruess, R.W.; Hendrick, R.L. Fine root architecture of nine North American trees. Ecol. Monogr. 2002, 72, 293-309.
- Xia, M.X.; Guo, D.L.; Pregitzer, K.S. Ephemeral root modules in Fraxinus mandshurica. New Phytol. 2010, 188, 1065-1074.
- Kong, D.; Ma, C.; Zhang, Q.; Li, L.; Chen, X.; Zeng, H.; Guo, D. Leading dimensions in absorptive root trait variation across 96 subtropical forest species. New Phytol. 2014, 203, 863–872.
- Clarify if the soil source (greenhouse) is representative of the actual urban soil where these trees are planted. Urban trees are often exposed to varied soil conditions, such as sandy or ex-constructed soil with a high load of coarse particles.
Response: Thanks for the reviewer’s suggestion.
Your suggestions are very valuable and helpful to our paper. As you mentioned, urban trees are often exposed to varied soil conditions, such as sandy or ex-constructed soil with a high load of coarse particles. Therefore, in our study, the soil we selected was the surface soil from the artificial forests of Pinus tabuliformis and Fraxinus chinensis at Nankai University, which was the sandy loam with low clay content and relatively resource-poor [1,2]. Campus green space (Nankai University) belongs to the attached green space, which is one of the important parts of urban green space [3]. We took topsoil from the campus and mixed it homogeneously and filled each pot with three-quarters of the soil in the greenhouse, which could represent the actual urban soil where these trees were planted. Thanks for your advice again, we have added the soil sources in the M&M based on your comments (See L109-111 in the revised manuscript).
- Foti, L.; Barot, S.; Gignoux, J.; Grimaldi, M.; Lata, J.; Lerch, T.Z.; Nold, F.; Nunan, N.; Raynaud, X.; Abbadie, L.; et al. Topsoil characteristics of forests and lawns along an urban–rural gradient in the Paris region (France). Soil Use Manage. 2020, 37, 749-761.
- Lan, T.; Guo, S.; Han, J.; Yang, Y.; Zhang, K.; Zhang, Q.; Yang, W.; Li, P. Evaluation of physical properties of typical urban green space soils in Binhai Area, Tianjin, China. Urban For. Urban Gree. 2019, 44, 126430.
- Li, X.; Ni, G.; Dewancker, B.J. Improving the attractiveness and accessibility of campus green space for developing a sustainable university environment. Environ. Sci. Pollut. R. 2019, 26, 33399-33415.
- Include units on the X axes of each graph for clarity.
Response: Thanks for the reviewer’s suggestion.
We apologize for the oversight regarding the graphs. We have added the units in the figures to ensure the clarity (See Figures in the revised manuscript).
Discussion:
- Explore and discuss the physiological mechanisms within the trees that affect the responses to varying N concentrations in the soil.
Response: Thanks for the reviewer’s suggestion.
Your suggestion is of great importance to us. The discussion about the physiological mechanisms within the trees that affect the responses to varying N concentrations in the soil have been added in the discussion (See Discussion in the revised manuscript).
We explored the physiological mechanism mainly in terms of the response of tree functional traits to N addition. First, we discussed the changes of plant photosynthetic mechanism according to the variation of LNC and SLA. We also measured the leaf traits like LDMC and LTD related to nutrient storage and discussed the difference under district N addition conditions. As for roots traits, we considered the root nutrient uptake and storage and respiratory metabolism by the measurement of root system architecture (SRL, SRA, RTD) and RNC. Meanwhile, we explore the response of stoichiometric ratios (C/N ratios) within plant aboveground and belowground tissues to environmental change, which could affect the synthesis of biomacromolecule (such as protein) and other physiological process (See 4.2. and 4.3. in the revised manuscript).
- Consider including simple observations of plant tissue, in addition to weight measurements. This information could provide insights into whether changes in biomass and R/S biomass partitioning correlate with plant anatomy.
Response: Thanks for the reviewer’s suggestion.
We are sorry for not consideration the plant anatomy traits in our experiment. As you mentioned, plant tissue is very important and it’s a good predictor to explore the plant growth strategy in response to the environmental changes [1]. However, we had ignored this question at that time, which could not fully explain the response mechanism of the two tree species. Therefore, we have added to explain the question into the Limitation of our manuscript (See L433-439 in the revised manuscript). Furthermore, our subsequent experiments will be conducted at forest farms in natural forest ecosystem. We will consider to measure the relatively anatomy traits like xylem vessel and leaf stomatal traits of the mature tree species to verify this study.
- Castagneri, D.; Regev, L.; Boaretto, E.; Carrer, M. Xylem anatomical traits reveal different strategies of two Mediterranean oaks to cope with drought and warming. Environ. Exp. Bot. 2017, 133, 128-138.
Limitations and Future Directions:
- Add a section discussing the limitations of the study and suggesting potential future directions. This will enhance the manuscript's significance by acknowledging constraints and proposing avenues for further research.
Response: Thanks for the reviewer’s suggestion.
Your advice is very valuable to us. we have added the “Limitations and Future Directions” in our manuscript according to your suggestions (See L423-439 in the revised manuscript). As you mentioned, this will make the structure of the manuscript more reasonable and complete, which also enhance the manuscript's significance.
In the Limitations and Future Directions, we have discussed the two main issues. First, the tree species we selected were the seedlings for a short-term simulation experiment, and the experimental site was in the greenhouse. It could not fully represent those of the mature trees in the natural forest ecosystem. Second, as you pointed out in Q9, we didn’t measure the plant anatomical traits in our study, which could be related to tree biomass allocation. Therefore, it’s necessary to explore the response of mature conifers and broadleaved tree species to N deposition at a longer time scale. Meanwhile, we also should measure the both economic and anatomical traits of different tree species, so as to fully understand the response mechanism of tree biomass allocation.
By addressing these suggestions, the manuscript can be refined for improved clarity, consistency, and depth of analysis.

Reviewer 2 Report
Comments and Suggestions for Authors
This paper presents data on the reaction of Pinus tabuliformis and Fraxinus chinensi to N addition to soil. This study has some scientific and practical points. However, there are several questions and remarks regarding the paper by Zhang et al. “Urban dominant trees followed the optimal partitioning theory and increased root biomass allocation and nutrient uptake under elevated nitrogen deposition”.
The main goal of this study is unclear. Why compare woody plant species that are evolutionarily distinct from one another (Gymnosperms vs. Angiosperms)? How can you compare plants that have distinct growth and development strategies? Different plant species that have diverged over evolution will react differently to nitrogen inputs. Without such extensive investigations, the conclusion that the response of various tree species to nitrogen inputs differed is obvious (“…the responses … were different, due to species specificity.” (Lines 421-422)). Why were no two closely related tree species chosen for comparison? Consider comparing two species of ash or two species of pine, for instance.
Where did you receive the seeds that were used to grow the experimental plants?
Soil used for plant sprouting was extracted from a depth of 0–1 metres. Given that the physical and chemical characteristics of soil considerably alter with depth, why did you take the soil so deep?
“The amount of N addition was based on the total amount of N deposition and its triple in Tianjin.” (Lines 123-124) In Tianjin? Reference should be made here to the data source.
Figure 5. Why does for ash not perform the same analysis that for pine does?
Author Response
Independent Review Report, Reviewer 2
This paper presents data on the reaction of Pinus tabuliformis and Fraxinus chinensis to N addition to soil. This study has some scientific and practical points. However, there are several questions and remarks regarding the paper by Zhang et al. “Urban dominant trees followed the optimal partitioning theory and increased root biomass allocation and nutrient uptake under elevated nitrogen deposition”.
General comments:
- The main goal of this study is unclear. Why compare woody plant species that are evolutionarily distinct from one another (Gymnosperms vs. Angiosperms)? How can you compare plants that have distinct growth and development strategies? Different plant species that have diverged over evolution will react differently to nitrogen inputs. Without such extensive investigations, the conclusion that the response of various tree species to nitrogen inputs differed is obvious (“…the responses … were different, due to species specificity.” (Lines 421-422)). Why were no two closely related tree species chosen for comparison? Consider comparing two species of ash or two species of pine, for instance.
Response: Thanks for the reviewer’s suggestion.
The question you raised about the selection of tree species is very important for us. There are two main reasons why we decided to selected the two tree species. First, Pinus tabuliformis and Fraxinus chinensis, as the dominated tree species in urban green space in North China, play the important roles in alleviating urban environmental pressure and providing ecological and economic benefits. With the intensification of atmospheric N deposition, the growth and development strategies of the two species will certainly change. Therefore, it is necessary to explore the effects of N inputs on tree biomass patterns and functional traits, so as to fully understand the response mechanisms of urban greening tree species to environmental changes, and better protect and manage the urban green space.
Second, according to taxonomy, P. tabuliformis is coniferous, which belongs to gymnosperm, while F. chinensis is a broadleaved tree species of angiosperms. As you mentioned, the two trees have distinct growth strategies, functional traits, and response to environmental change, so that we could speculate the response of various tree species to N inputs might be different. However, our study is not just about comparing the differences between the two species. The main purpose is to explore the specific response mechanism of different types of tree species to N inputs, and how they modify their aboveground and belowground resource acquisition strategies and biomass allocation patterns to adapt to high N deposition environment. It is because of the phylogenetic differences between the two species that we chose them to further explore the different biomass allocation and functional traits, which could better understand the growth strategies of different coniferous and broadleaved tree species in response to environmental change. We believe it is very meaningful for the selection and specialized conservation of different tree species in urban green spaces. In addition, our experiment design is referred to the previous studies on the response of evolutionarily different tree species to environmental changes [1-4].
Overall, according to your question, we added the reason why we selected the two tree species in the M & M to make the goal clearer (See L98-103 in the revised manuscript).
In our conclusion, the original sentence (The results suggested that the responses of conifers and broadleaf trees to N deposition were different, due to species specificity) was a general explanation of the results about our study, and we have explained the response mechanism of different tree species to N addition in the later part. To not confuse the readers about the expression, we have deleted the sentence and revised the conclusion (See L441-442 in the revised manuscript).
- Wang, C.; McCormack, M.L.; Guo, D.; Li, J. Global meta‐analysis reveals different patterns of root tip adjustments by angiosperm and gymnosperm trees in response to environmental gradients. J. Biogeogr. 2018, 46, 123-133.
- Hu, Y.; Yu, W.; Cui, B.; Chen, Y.; Zheng, H.; Wang, X. Pavement Overrides the Effects of Tree Species on Soil Bacterial Communities. Int. J. Environ. Res. Public Health 2021, 18, 2168
- Zhang, Q.; Zhang, J.; Shi, Z.; Kang, B.; Tu, H.; Zhu, J.; Li, H. Nitrogen addition and drought affect nitrogen uptake patterns and biomass production of four urban greening tree species in North China. Sci. Total Environ. 2023, 893, 164893.
- Hu, X.; Wang, X.; Abbas, T.; Fang, T.; Miao, D.; Li, Y.; Chang, S.X.; Li, Y. Higher ammonium-to-nitrate ratio shapes distinct soil nitrifying community and favors the growth of Moso bamboo in contrast to broadleaf tree species. Biol. Fert. Soils 2021, 57, 1171-1182.
- Where did you receive the seeds that were used to grow the experimental plants?
Response: Thanks for the reviewer’s suggestion.
In our study, we purchased and used the healthy bare-rooted seedings (1-year-old) of the two common tree species, instead of the seeds (See L105-107 in the revised manuscript). In general, the seedlings selected in the experiment should preferably be germinated from seeds planted in the greenhouse, and were collected in autumn or spring from the urban green space in Tianjin Province [1,2]. However, since our overall experimental cycle is relatively short and the seed germination process takes at least a year, we directly choose to purchase 1-year-old seedlings of tree species rather than seeds that require germination cycles. The healthy seedlings of the two tree species were provided by Dianzhuang Seeding Company, Jiangsu, China, and each seedings with a similar size (about 50 cm height). The seeds germination was companied at the nursery garden, and the mature seeds were collected from healthy trees species of Pinus tabuliformis and Fraxinus chinensis in their own nursery.
- Soil used for plant sprouting was extracted from a depth of 0–1 metres. Given that the physical and chemical characteristics of soil considerably alter with depth, why did you take the soil so deep?
Response: Thanks for the reviewer’s suggestion.
We are sorry for the incorrect description of the soil. In our experiment, the soil that we actually dug up was 0-20cm of topsoil, which was a reference to Hu's study [1]. Because the surface soil is rich in soil organic matter, which contributes to improved soil air diffusion, infiltration, water holding capacity and aggregate stability and is suitable for planting plants [2-4]. Therefore, we used topsoil for plant growth, which helped to improve the availability of soil nutrients and water, and facilitates plant growth and development. Sorry again for our mistake, we have revised the description of the soil depth (See L109-110 in the revised manuscript).
- Hu, X.; Wang, X.; Abbas, T.; Fang, T.; Miao, D.; Li, Y.; Chang, S.X.; Li, Y. Higher ammonium-to-nitrate ratio shapes distinct soil nitrifying community and favors the growth of Moso bamboo in contrast to broadleaf tree species. Biol. Fert. Soils 2021, 57, 1171-1182.
- Blanco‐Canqui, H.; Lal, R. Corn stover removal impacts on micro-scale soil physical properties. Geoderma 2008, 145, 335-346.
- Smith, W.N.; Grant, B.B.; Campbell, C.A.; McConkey, B.; Desjardins, R.L.; Kröbel, R.; Malhi, S.S. Crop residue removal effects on soil carbon: Measured and inter-model comparisons. Agr. Ecosyst. Environ. 2012, 161, 27-38.
- Li, S.; Gu, X.; Zhuang, J.; An, T.; Pei, J.; Xie, H.; Li, H.; Fu, S.; Wang, J. Distribution and storage of crop residue carbon in aggregates and its contribution to organic carbon of soil with low fertility. Soil Till. Res. 2016, 155, 199-206.
- “The amount of N addition was based on the total amount of N deposition and its triple in Tianjin.” (Lines 123-124) In Tianjin? Reference should be made here to the data source.
Response: Thanks for the reviewer’s suggestion.
We are sorry for not writing the reference for the amount of N addition in our experiment. Our experimental design was based on the level of N deposition in Tianjin and the amount of N added in some previous studies [1-3]. Since the total amount of N deposition in Tianjin was reported as 3.5 g N m-2 year−1 [3], our low N addition amount was 3.5 g N m-2 year−1 to simulate the level of N deposition in Tianjin. In addition, to simulate the possible environment of high N deposition in the future, we selected three times the level of N deposition in Tianjin (10.5 g N m-2 year−1) to explore it impact on growth patterns of tree species. In some cities in southern China, the amount of N deposition is more than 10 g N m-2 year−1 [4]. Therefore, it is reasonable for us to choose three times the amount of N deposition for simulation experiment. We have added references in the M & M to make it more comprehensive for readers to understand (See L121-122 in the revised manuscript).
- Zhang, Q.; Zhang, J.; Shi, Z.; Kang, B.; Tu, H.; Zhu, J.; Li, H. Nitrogen addition and drought affect nitrogen uptake patterns and biomass production of four urban greening tree species in North China. Sci. Total Environ. 2023, 893, 164893.
- Zhang, Q.; Hao, G.; Li, M.; Li, L.; Kang, B.; Yang, N.; Li, H. Transformation of plant to resource acquisition under high nitrogen addition will reduce green roof ecosystem functioning. Front. Plant Sci. 2022, 13, 894782.
- Yu, G.; Jia, Y.; He, N.; Zhu, J.; Chen, Z.; Wang, Q.; Piao, S.; Liu, X.; He, H.; Guo, X.; et al. Stabilization of atmospheric nitrogen deposition in China over the past decade. Nat. Geosci. 2019, 12, 424-429.
- Huang, P.; Shen, F.; Abbas, A.; Wang, H.; Du, Y.; Du, D.; Hussain, S.; Javed, T.; Alamri, S.A. Effects of different nitrogen forms and competitive treatments on the growth and antioxidant system of Wedelia trilobata and Wedelia chinensis under high nitrogen concentrations. Front. Plant Sci. 2022, 13, 851099.
- Figure 5. Why does for ash not perform the same analysis that for pine does?
Response: Thanks for the reviewer’s suggestion.
We think this question you raised is very important for us. According to our Pearson analysis (See Figure S1 in the Supplementary Materials), there were significant correlations between aboveground and belowground functional traits of Pinus tabuliformis (Figure S1a), so the functional traits with significance were selected for linear regression analysis (See Figure 5 in the revised manuscript). However, we did not perform the same linear regression analysis for Fraxinus chinensis in the original manuscript because the correlated heat map did not significantly correlate aboveground and belowground functional traits (Figure S1b). Meanwhile, we also explained it in the 3.2. Aboveground and Belowground Functional Traits of the Results (See L240-241 in the revised manuscript). Nevertheless, to make the analysis of the article more complete and easier to understand, we have taken your suggestion to carry out the same linear regression analysis for the Fraxinus chinensis in the Supplementary materials and added additional explanations in the manuscript (See L240-242 and Figure S2 in the revised manuscript and Supplementary materials).

Round 2
Reviewer 2 Report
Comments and Suggestions for Authors
I appreciate that the authors have considered most of the comments. The manuscript have been carefully revised.